# GENDIS: Genetic Discovery of Shapelets

**DOI:** 10.3390/s21041059

**Published:** 2021-02-04

**Authors:** Gilles Vandewiele, Femke Ongenae, Filip De Turck

**Affiliations:** IDLab, Ghent University—imec, 9052 Ghent, Belgium; femke.ongenae@ugent.be (F.O.); filip.deturck@ugent.be (F.D.T.)

**Keywords:** genetic algorithms, time series classification, time series analysis, explainable artificial intelligence (xAI), data mining

## Abstract

In the time series classification domain, shapelets are subsequences that are discriminative of a certain class. It has been shown that classifiers are able to achieve state-of-the-art results by taking the distances from the input time series to different discriminative shapelets as the input. Additionally, these shapelets can be visualized and thus possess an interpretable characteristic, making them appealing in critical domains, where longitudinal data are ubiquitous. In this study, a new paradigm for shapelet discovery is proposed, which is based on evolutionary computation. The advantages of the proposed approach are that: (i) it is gradient-free, which could allow escaping from local optima more easily and supports non-differentiable objectives; (ii) no brute-force search is required, making the algorithm scalable; (iii) the total amount of shapelets and the length of each of these shapelets are evolved jointly with the shapelets themselves, alleviating the need to specify this beforehand; (iv) entire sets are evaluated at once as opposed to single shapelets, which results in smaller final sets with fewer similar shapelets that result in similar predictive performances; and (v) the discovered shapelets do not need to be a subsequence of the input time series. We present the results of the experiments, which validate the enumerated advantages.

## 1. Introduction

### 1.1. Background

Due to the rise of the Internet-of-Things (IoT), a mass adoption of sensors in all domains, including critical domains such as health care, can be noted. These sensors produce data of a longitudinal form, i.e., time series. Time series differ from classical tabular data, since a temporal dependency is present where each value in the time series correlates with its neighboring values. One important task that emerges from this type of data is the classification of time series in their entirety. A model able to solve such a task can be applied in a wide variety of applications, such as distinguishing between normal brain activity and epileptic activity [1], determining different types of physical activity [2], or profiling electronic appliance usage in smart homes [3]. Often, the largest discriminative power can be found in smaller subsequences of these time series, called shapelets. Shapelets semantically represent intelligence on how to discriminate between the different targets of a time series dataset [4]. We can use a set of shapelets and the corresponding distances from each of these shapelets to each of the input time series as features for a classifier. It has been shown that such an approach outperforms a nearest neighbor search based on dynamic time warping distance on almost every dataset, which has been deemed to be the state-of-the-art for a long time [5]. Moreover, shapelets possess an interpretable characteristic since they can easily be visualized and retraced back to the input signal, making them very interesting for decision support applications in critical domains, such as the medical domain. In these critical domains, it is of vital importance that a corresponding explanation can be provided alongside a prediction, since the wrong decision can have a significant negative impact.

### 1.2. Related Work

Shapelet discovery was initially proposed by Ye and Keogh [6]. Unfortunately, the initial algorithm quickly becomes intractable, even for smaller datasets, because of its large computational complexity $O(N^{2}M^{4})$, with *N* the number of time series and *M* the length of the smallest time series in the dataset). This complexity was improved two years later, when Mueen et al. [7] proposed an extension to this algorithm that makes use of caching for faster distance computation and a better upper bound for candidate pruning. These improvements reduce the complexity to $O(N^{2}M^{3})$, but have a larger memory footprint. Rakthanmanon et al. [8] proposed an approximative algorithm, called Fast Shapelets (fs), that finds a suboptimal shapelet in $O(NM^{2})$ by first transforming the time series in the original set to Symbolic Aggregate approXimation (sax) representations [9]. Although no guarantee can be made that the discovered shapelet is the one that maximizes a pre-defined metric, they showed that they are able to achieve very similar classification performances, empirically on 32 datasets.

All the aforementioned techniques search for a single shapelet that optimizes a certain metric, such as information gain. Often, one shapelet is not enough to achieve good predictive performances, especially for multi-class classification problems. Therefore, the shapelet discovery is applied in a recursive fashion in order to construct a decision tree. Lines et al. [10] proposed Shapelet Transform (st), which performs only a single pass through the time series dataset and maintains an ordered list of shapelet candidates, ranked by a metric, and then finally takes the top-k from this list in order to construct features. While the algorithm only performs a single pass, the computational complexity still remains $O(N^{2}M^{4})$, which makes the technique intractable for larger datasets. Extensions to this technique have been proposed in the subsequent years, which drastically improved the performance of the technique [11,12]. Lines et al. [10] compared their technique to 36 other algorithms for time series classification on 85 datasets [13], which showed that their technique is one of the top-performing algorithms for time series classification and the best-performing shapelet extraction technique in terms of predictive performance.

Grabocka et al. [4] proposed a technique where shapelets are learned through gradient descent, in which the linear separability of the classes after transformation to the distance space is optimized, called Learning Time Series Shapelets (lts). The technique is competitive with st, while not requiring a brute-force search, making it tractable for larger datasets. Unfortunately, lts requires the user to specify the number of shapelets and the length of each of these shapelets, which can result in a rather time-intensive hyper-parameter tuning process in order to achieve a good predictive performance. Three extensions of lts, which improve the computational runtime of the algorithm, have been proposed in the subsequent years. Unfortunately, in order to achieve these speedups, predictive performance had to be sacrificed. A first extension is called Scalable Discovery (sd) [14]. It is the fastest of the three extensions, improving the runtime by two to three orders of magnitude, but at the cost of having a worse predictive performance than lts on almost every tested dataset. Second, in 2015, Ultra-Fast Shapelets (ufs) [15] was proposed. It is a better compromise of runtime and predictive performance, as it is an order of magnitude slower than sd, but sacrifices less of its predictive performance. The final and most recent extension is called the Fused LAsso Generalized eigenvector method (flag) [16]. It is the most notable of the three extensions as it has runtimes competitive with sd, while being only slightly worse than lts in terms of predictive performance.

Several enhancements to shapelet discovery have recently been investigated as well. Wang et al. [17] investigated adversarial regularization in order to enhance the interpretability of the discovered shapelets. Guillemé et al. [18] investigated the added value of the location information of the discovered shapelets on top of distance-based information.

Most of the prior work regarding shapelet discovery was performed using univariate data. However, many real-world datasets are multi-variate. Extending shapelet discovery algorithm to deal with multi-variate data has therefore been gaining increasing interest in the time series analysis domain. Most of the existing works extend the gradient-based framework of lts [19,20] or perform a brute-force search with sampling [21].

### 1.3. Our Contribution

This paper is the first to investigate the feasibility of an evolutionary algorithm in order to discover a set of shapelets from a collection of labeled time series. The aim of the proposed algorithm, GENetic DIscovery of Shapelets (gendis), is to achieve state-of-the-art predictive performances similar to the best-performing algorithm, st, with a smaller number of shapelets, while having a low computational complexity similar to lts.

gendis tries to retain as many of the positive properties from lts as possible such as its scalable computational complexity, the fact that entire sets of shapelets are discovered as opposed to single shapelets, and that it can discover shapelets outside the original dataset. We demonstrate the added value of these two final properties through intuitive experiments in Section 3.2 and Section 3.3, respectively. Moreover, gendis has some benefits over lts. First, genetic algorithms are gradient-free, allowing for any objective function and an easier escape from local optima. Second, the total amount of shapelets and the length of each of these shapelets do not need to be defined prior to the discovery, alleviating the need to tune these, which could be computationally expensive and may require domain knowledge. Finally, we show by a thorough comparison, in Section 3.5, that gendis empirically outperforms lts in terms of predictive performance.

## 2. Materials and Methods

We first explain some general concepts from the time series analysis and shapelet discovery domain, on which we will then build further to elaborate our proposed algorithm, gendis.

### 2.1. Time Series Matrix and Label Vector

The input to a shapelet discovery algorithm is a collection of *N* time series. For the ease of notation, we assume that the time series are synchronized and have a fixed length of *M*, resulting in an input matrix $T\in R^{N\times M}$. It is important to note that gendis could perfectly work with variable length time series as well. In that case, *M* would be equal to the minimal time series length in the collection. Since shapelet discovery is a supervised approach, we also require a label vector y of length *N*, with each element $y_{i}\in \left\{1,\ldots ,C\right\}$ with *C* the number of classes and $y_{i}$ corresponding to the label of the *i*-th time series in T.

### 2.2. Shapelets and Shapelet Sets

Shapelets are small time series that semantically represent intelligence on how to discriminate between the different targets of a time series dataset. In other words, they are very similar to subsequences from time series of certain (groups of) classes, while being dissimilar to subsequences of time series of other classes. The output of a shapelet discovery algorithm is a collection of *K* shapelets, $S\;=\;\{s_{1},\ldots ,s_{K}\}$, called a shapelet set. In gendis, *K* and the length of each shapelet do not need to be defined beforehand, and each shapelet can have a variable length, smaller than *M*. These *K* shapelets can then be used to extract features for the time series, as we will explain subsequently.

### 2.3. Distance Matrix Calculation

Given an input matrix T and a shapelet set S, we can construct a pairwise distance matrix $D\in R^{N\times K}$:(1)dist(S,T)=D

The distance matrix, D, is constructed by calculating the distance between each (*t*, *s*) pair, where t∈T is an input time series and s∈S a shapelet from the candidate shapelet set. This matrix can then be fed to a machine learning classifier. Often, K<<M, such that we effectively reduce the dimension of our data. In order to calculate the distance from a shapelet *s* in S to a time series *t* from T, we slide the shapelet across the time series and take the minimum distance:(2)$$dist\left(s,t\right)=\min_{1\;\leq \;i\;\leq \;|t|-|s|}d(s,t[i:i+\left|s\right|-1])$$
with d(.) a distance metric, such as the Euclidean distance, and t[i:i+|s|−1] a slice from *t* starting at index *i* and having the same length as *s*.

### 2.4. Shapelet Set Discovery

A conceptual overview of a shapelet discovery algorithm is depicted in Figure 1. The discovery algorithm tries to find a set of shapelets, S, that produces a distance matrix, D, that minimizes the loss function, L, of the machine learning technique to which it is fed, h(.), given the ground truth, y.
(3)$$\min_{S}L\left(h\left(dist\left(T,S\right)\right),y\right)$$

Once shapelets are found, these can be used to transform the time series into features that correspond to distances from each of the time series to the shapelets in the set. These features can then be fed to a classifier. It should be noted that both the shapelet discovery and the classification component can be trained jointly end-to-end. However, in gendis, these components are decoupled.

### 2.5. Genetic Discovery of Interpretable Shapelets

In this paper, we propose a genetic algorithm that evolves a set of variable-length shapelets, S, in $O(NM^{2})$, which produces a distance matrix D, based on a collection of time series T that results in optimal predictive performance when provided to a machine learning classifier. The intuition behind the approach is similar to lts, which we mentioned in Section 1.2, but the advantage is that both the size of S (*K*) and the length of each shapelet s∈S are evolved jointly, alleviating the need to specify the number of shapelets and the length of each shapelet prior to the extraction. Moreover, the technique is gradient-free, which allows for non-differentiable objectives and escaping local optima more easily.

#### 2.5.1. Conceptual Overview

The building blocks of a genetic algorithm consist of at least a crossover, mutation, and selection operator [22]. Additionally, we seed, or initialize, the algorithm with specific candidates instead of completely random candidates [23] and apply elitism [24] to make sure the fittest candidate set is never discarded from the population or never experiences mutations that detriment its fitness.

A conceptual overview of gendis is provided in Figure 2. Initially, a population is constructed through initialization operators. Afterwards, the individuals in the population are evolved iteratively to increase their quality. An iteration, or generation, of the algorithm first consists of calculating the fitness values for new individuals. When all fitness values are known, tournament selection is applied to select pairs of individuals or candidates. These pairs then undergo a crossover to generate new offspring, which are added to the population. Then, each of the individuals in the population undergoes mutations with a certain probability. Finally, the fittest individuals are selected from the population, and this process repeats until convergence or until the stop criteria are met. Each of these operations are elaborated upon in the following subsections.

#### 2.5.2. Initialization

In order to seed the algorithm with initial candidate sets, we generate *P* candidate sets $S^{{}^{\prime }}$ containing *K* shapelets, with *K* a random integer picked uniformly from [2,W], *W* a hyper-parameter of the algorithm, and *P* the population size. *K* is randomly chosen for each individual, and the default value of *W* is set to be $\sqrt{M}$. These two boundaries are chosen to be low in order to start with smaller candidate sets and grow them incrementally. This is beneficial for both the size of the final shapelet set, as well as the runtime of each generation. For each candidate set we initialize, we randomly pick one of the following two strategies with equal probability:
**Initialization** **1.** Apply K-means on a set of random subseries of a fixed random length sampled from T. The K resulting centroids form a candidate set.
**Initialization** **2.** Generate K candidates of random lengths (∈{4,…,max_len}) by sampling them from T.

max_len is a hyper-parameter that limits the length of the discovered shapelets, in order to combat overfitting. While Initialization 1 results in strong initial individuals, Initialization 2 is included in order to increase the population diversity and to decrease the time required to initialize the entire population.

#### 2.5.3. Fitness

One of the most important components of a genetic algorithm is its fitness function. In order to determine the fitness of a candidate set $S^{{}^{\prime }}$, we first construct $D^{{}^{\prime }}$, which is the distance matrix obtained by calculating the distances between $S^{{}^{\prime }}$ and T. The goal of our genetic algorithm is to find an $S^{{}^{\prime }}$ that produces a $D^{{}^{\prime }}$ that results in the most optimal predictive performance when provided to a classifier. We measure the predictive performance directly by means of an error function defined on the predictions of a logistic regression model and the provided label vector y. When two candidate shapelet sets produce the same error, the set with the lowest complexity is deemed to be the fittest. The complexity of a shapelet set is expressed as the sum of shapelet lengths ($\sum_{s\in S}\left|s\right|$).

The fitness calculation is the bottleneck of the algorithm. Calculating the distance of a shapelet with length *L* to a time series of length *M* requires (M−L+1)×L pointwise comparisons. Thus, in the worst case, $O(M^{2})$ operations need to be performed per time series, resulting in a computational complexity of $O(NM^{2})$. We apply these distance calculations to each individual representing a collection of shapelets from our population, in each generation. Therefore, the complexity of the entire algorithm is equal to $O(GPKNM^{2})$, with *G* the total number of generations, *P* the population size, and *K* the (maximum) number of shapelets in the bag each individual of the population represents.

#### 2.5.4. Crossover

We define three different crossover operations, which take two candidate shapelet sets, $S^{{}^{\prime }}$ and $S^{{}^{\prime \prime }}$, as the input and produce two new sets, $S^{*}$ and $S^{**}$:
**Crossover** **1.** Apply one- or two-point crossover on two shapelet sets (each with a probability of 50%). In other words, we create two new shapelet sets that are composed of shapelets from both $S^{{}^{\prime }}$ and $S^{{}^{\prime \prime }}$. An example of this operation is provided in Figure 3.
**Crossover** **2.** Iterate over each shapelet s in $S^{{}^{\prime }}$, and apply one- or two-point crossover (again with a probability of 50%) with another randomly chosen shapelet from $S^{{}^{\prime \prime }}$ to create $S^{*}$. Apply the same, vice versa, to obtain $S^{**}$. This differs from the first crossover operation as the one- or two-point crossover is performed on individual shapelets as opposed to entire sets. An example of this operation can be seen in Figure 4.
**Crossover** **3.** Iterate over each shapelet s in $S^{{}^{\prime }}$, and merge it with another randomly chosen shapelet from $S^{{}^{\prime \prime }}$. The merging of two shapelets can be done by calculating the mean (or barycenter) of the two time series. When two shapelets being merged have varying lengths, we merge the shorter shapelet with a random part of the longer shapelet. A schematic overview of this strategy, on shapelets having the same length, is depicted in Figure 5.

It is possible that all or no techniques are applied on a pair of individuals. Each technique has a probability equal to the configured crossover probability ($p_{\mathrm{crossover}}$) of being applied.

#### 2.5.5. Mutations

The mutation operators are a vital part of the genetic algorithm, as they ensure population diversity and allow escaping from local optima in the search space. They take a candidate set $S^{{}^{\prime }}$ as the input and produce a new, modified $S^{*}$. In our approach, we define three simple mutation operators:
**Mutation** **1.** Take a random $s\in S^{{}^{\prime }}$, and randomly remove a variable amount of data points from the beginning or ending of the time series.
**Mutation** **2.** Remove a random $s\in S^{{}^{\prime }}$.
**Mutation** **3.** Create a new candidate using Initialization 2, and add it to $S^{{}^{\prime }}$.

Again, all techniques can be applied on a single individual, each having a probability equal to the configured mutation probability ($p_{\mathrm{mutation}}$).

#### 2.5.6. Selection, Elitism, and Early Stopping

After each generation, a fixed number of candidate sets is chosen based on their fitness for the next generation. Many different techniques exist to select these candidate sets. We chose to apply tournament selection with small tournament sizes. In this strategy, a number of candidate sets is sampled uniformly from the entire population to form a tournament. Afterwards, one candidate set is sampled from the tournament, where the probability of being sampled is determined by its fitness. Smaller tournament sizes ensure better population diversity as the probability of the fittest individual being included in the tournament decreases. Using this strategy, it is however possible that the fittest candidate set from the population is never chosen to compete in a tournament. Therefore, we apply elitism and guarantee that the fittest candidate set is always transferred to the next generation’s population. Finally, since it can be hard to determine the ideal number of generations that a genetic algorithm should run, we implemented early stopping where the algorithm preemptively stops as soon as no candidate set with a better fitness has been found for a certain number of iterations (patience).

#### 2.5.7. List of All Hyper-Parameters

We now present an overview of all hyper-parameters included in gendis, along with their corresponding explanation and default values.
Maximum shapelets per candidate (*W*): the maximum number of shapelets in a newly generated individual during initialization (default: $\sqrt{M}$).Population size (*P*): the total number of candidates that are evaluated and evolved in every iteration (default: 100).Maximum number of generations (*G*): the maximum number of iterations the algorithm runs (default: 100).Early stopping patience (patience): the algorithm preemptively stops evolving when no better individual has been found for patience iterations (default: 10).Mutation probability ($p_{\mathrm{mutation}}$): the probability that a mutation operator gets applied to an individual in each iteration (default: 0.1).Crossover probability ($p_{\mathrm{crossover}}$): the probability that a crossover operator is applied on a pair of individuals in each iteration (default: 0.4).Maximum shapelet length (max_len): the maximum length of the shapelets in each shapelet set (individual) (default: *M*).The operations used during the initialization, crossover, and mutation phases are configurable as well (default: all mentioned operations).

## 3. Results and Discussion

In the following subsections, we present the setup of different experiments and the corresponding results in order to highlight the advantages of gendis.

### 3.1. Efficiency of Genetic Operators

In this section, we assess the efficiency of the introduced genetic operators by evaluating the fitness as a function of the number of generations using different sets of operators. It should be noted that our implementation easily allows configuring the number and type of operators used for each of the different steps in the genetic algorithm, allowing the user to tune these according to the dataset.

#### 3.1.1. Datasets

We picked six datasets, with varying characteristics, to evaluate the fitness of different configurations. The chosen datasets and their corresponding properties are summarized in Table 1.

#### 3.1.2. Initialization Operators

We first compare the fitness of GENDIS using three different sets of initialization operators:Initializing the individuals with K-means (Initialization 1)Randomly initializing the shapelet sets (Initialization 2)Using both initialization operations

Each configuration was tested using a small population (25 individuals), in order to reduce the required computational time, for 75 generations, as the impact of the initialization is highest in the earlier generations. All mutation and crossover operators were used. We show the average fitness of all individuals in the population in Figure 6. From these results, we can conclude that the two initialization operators are competitive with each other, as one operator will outperform the other on several datasets and vice versa on the others.

#### 3.1.3. Crossover Operators

We now compare the average fitness of all individuals in the population, as a function of the number of generations, when configuring GENDIS to use four different sets of crossover operators:Using solely point crossovers on the shapelet sets (Crossover 1)Using solely point crossovers on individual shapelets (Crossover 2)Using solely merge crossovers (Crossover 3)Using all three crossover operations

Each run had a population of 25 individuals and ran for 200 generations. All mutation and initialization operators were used. As the average fitness is rather similar in the earlier generations, we truncated the first 50 measurements to better highlight the differences. The results are presented in Figure 7. As can be seen, it is again difficult to single out an operation that significantly outperforms the others.

#### 3.1.4. Mutation Operators

The same experiment was performed to assess the efficiency of the mutation operators. Four different configurations were used:Masking a random part of a shapelet (Mutation 1)Removing a random shapelet from the set (Mutation 2)Adding a shapelet, randomly sampled from the data, to the set (Mutation 3)Using all three mutation operations

The average fitness of the individuals, as a function of the number of generations, is depicted in Figure 8. It is clear that the addition of shapelets (Mutation 3) is the most significant operator. Without it, the fitness quickly converges to a sub-optimal value. The removal and masking of shapelets does not seem to increase the average fitness often, but they are important operators in order to keep the the number of shapelets and the length of the shapelets small.

### 3.2. Evaluating Sets of Candidates Versus Single Candidates

A key factor of gendis is that it evaluates entire sets of shapelets (a dependency between the shapelets is introduced), as opposed to evaluating single candidates independently and taking the top-k. The disadvantage of the latter approach is that similar shapelets will achieve similar values given a certain metric. When entire sets are evaluated, we can optimize both the quality metric for candidate sets, as the size of each of these sets. This results in smaller sets with fewer similar shapelets. Moreover, interactions between shapelets can be explicitly taken into account. To demonstrate these advantages, we compare gendis to st, which evaluates candidate shapelets individually as opposed to shapelet sets, on an artificial three-class dataset, depicted in Figure 9. The constructed dataset contains a large number of very similar time series of Class 0, while having a smaller number of more dissimilar time series of Classes 1 and 2. The distribution of time series across the three classes in both the training and test dataset is thus skewed, with the number of samples in Classes 0,1,2 being equal to 25,5,5, respectively. This imbalance causes the independent approach to focus solely on extracting shapelets that can discriminate Class 0 from the other two, since the information gain will be highest for these individual shapelets. Clearly, this is not ideal as subsequences taken from the time series of Class 0 possess little to no discriminative power for the other two classes, as the distances to the time series from these two classes will be nearly equal.

We extracted two shapelets with both techniques, which allowed us to visualize the different test samples in a two-dimensional transformed distance space, as shown in Figure 10. Each axis of this space represents the distances to a certain shapelet. For the independent approach, we can clearly see that the distances of the samples for all three classes to both shapelets are clustered near the origin of the space, making it very hard for a classifier to draw a separation boundary. On the other hand, a very clear separation can be seen for the samples of the three classes when using the shapelets discovered by gendis, a dependent approach. The low discriminative power of the independent approach is confirmed by fitting a logistic regression model with a tuned regularization type and strength on the obtained distances. The classifier fitted on the distances extracted by the independent approach is only able to achieve an accuracy of 0.8286 ($\frac{29}{35}$) on the rather straight-forward dataset. The accuracy score of gendis, a dependent approach, equals 1.0.

### 3.3. Discovering Shapelets Outside the Data

Another advantage of gendis is that the discovery of shapelets is not limited to be a subseries from T. Due to the nature of the evolutionary process, the discovered shapelets can have a distance greater than zero to all time series in the dataset. More formally: ∃s∈S.∀t∈T.dist(s,t)>0. While this can be somewhat detrimental concerning interpretability, it can be necessary to get an excellent predictive performance. We demonstrate this through a very simple, artificial example. Assume we have a two-class classification problem and are provided two time series per class, as illustrated in Figure 11a. The extracted shapelet, and the distances to each time series, by a brute-force approach and a slightly modified version of gendis can be found in Figure 11b. The modification we made to gendis is that we specifically search for only one shapelet instead of an entire set of shapelets. We can see that the exhaustive search approach is not able to find a subseries in any of these four time series that separates both classes, while the shapelet extracted by gendis ensures perfect separation.

It is important to note here that discovering shapelets outside the data sacrifices interpretability for an increase in the predictive performance of the shapelets. As the operators that are used during the genetic algorithm are completely configurable for gendis, one can use only the first crossover operation (one- or two-point crossover on shapelet sets) to ensure all shapelets come from within the data.

### 3.4. Stability

In order to evaluate the stability of our algorithm, we compare the extracted shapelets of two different runs on the ItalyPowerDemand dataset. We set the algorithm to evolve a large population (100 individuals) for a large number of generations (500) in order to ensure convergence. Moreover, we limited the maximum number of extracted shapelets to 10, in order to keep the visualization clear. We then calculated the similarity of the discovered shapelets between the two runs, using dynamic time warping [25]. A heat map of the distances is depicted in Figure 12. While the discovered shapelets are not exactly the same, we can often find pairs that contain the same semantic intelligence, such as a saw pattern or straight lines.

### 3.5. Comparing gendis to fs, st, and lts

In this section, we compare our algorithm gendis to the results from Bagnall et al. [13], which are hosted online (www.timeseriesclassification.com). In that study, thirty-one different algorithms, including three shapelet discovery techniques, were compared on 85 datasets. The 85 datasets stem from different data sources and different domains, including electrocardiogram data from the medical domain and sensor data from the IoT domain. The three included shapelet techniques are Shapelet Transform (st) [10], Learning Time Series Shapelets (lts) [4], and Fast Shapelets (fs) [8]. A discussion of all three techniques can be found in Section 1.2.

For 84 of the 85 datasets, we conducted twelve measurements by concatenating the provided training and testing data and re-partitioning in a stratified manner, as done in the original study. Only the “Phoneme” dataset could not be included due to problems with downloading the data while executing this experiment. On every dataset, we used the same hyper-parameter configuration for gendis: a population size of 100, a maximum of 100 iterations, early stopping after 10 iterations, and crossover and mutation probabilities of 0.4 and 0.1, respectively. The only parameter that was tuned for every dataset separately was the maximum length for each shapelet, to combat overfitting. To tune this, we picked the length $l\in [\frac{M}{4},\frac{M}{2},\frac{3M}{4},M]$, which resulted in the best logarithmic (or entropy) loss using three-fold cross-validation on the training set. The distance matrix obtained through the extracted shapelets of gendis was then fed to a heterogeneous ensemble consisting of a rotation forest, random forest, support vector machine with a linear kernel, support vector with a quadratic kernel, and k-nearest neighbor classifier [26]. This ensemble matches the one used by the best-performing algorithm, st, closely. This is in contrast with fs, which produces a decision tree, and lts, which learns a separating hyperplane (similar to logistic regression) jointly with the shapelets. This setup is also depicted schematically in Figure 13. Trivially, the ensemble will empirically outperform each of the individual classifiers [27], but it does take a longer time to fit and somewhat takes the focus away from the quality of the extracted shapelets. Nevertheless, it is necessary to use an ensemble in order to allow for a fair comparison with st, as that was unfortunately used by Bagnall et al. [13] to generate their results. To give more insights into the quality of the extracted shapelets, we also report the accuracies using a Logistic Regression classifier. We tuned the type of regularization (ridge vs. lasso) and the regularization strength (C∈{0.001,0.01,0.1,1.0,10.0,100.0,1000.0}) using the training set. We recommend that future research compare their results to those obtained with the logistic regression classifier.

The mean accuracy over the twelve measurements of gendis in comparison to the mean of the hundred original measurements of the three other algorithms, retrieved from the online repository, can be found in Table 2 and Table 3. While a smaller number of measurements is conducted within this study, it should be noted that the measurements from Bagnall et al. [13] took over six months to generate. Moreover, accuracy is often not the most ideal metric to measure the predictive performance. Although it is one of the most intuitive metrics, it has several disadvantages such as skewness when the data are imbalanced. Nevertheless, the accuracy metric is the only one allowing for comparison to related work, as that metric was used in those studies. Moreover, the datasets used are merely benchmark datasets, and the goal is solely to compare the quality of the shapelets extracted by gendis to those of st. We recommend to use different performance metrics, which should be tailored to the specific use case. An example is using the area under the receiver operating characteristic curve (AUC) in combination with precision and recall for medical datasets.

For each dataset, we also performed an unpaired Student *t*-test with a cutoff value of 0.05 to detect statistically significant differences. When the performance of an algorithm for a certain dataset is statistically better than all others, it is indicated in bold. From these results, we can conclude that fs is inferior to the three other techniques, while st most often achieves the best performance, but at a very high computational complexity.

The average number of shapelets extracted by gendis is reported in the final column. The number of shapelets extracted by st in the original study equals 10×N. Thus, the total number of shapelets used to transform the original time series to distances is at least an order of magnitude less when using gendis. In order to compare the algorithms across all datasets, a Friedman ranking test [28] was applied with a Holm post-hoc correction [29,30]. We present the average rank of each algorithm using a critical difference diagram, with cliques formed using the results of the Friedman test with a Holm post-hoc correction at a significance cutoff level of 0.1 in Figure 14. The higher the cutoff level, the less probable it is to form cliques. For gendis, both the results obtained with the ensemble and with the logistic regression classifier are used. From this, we can conclude that there is no statistical difference between st and gendis, while both are statistically better than fs and lts.

## 4. Conclusions and Future Work

In this study, an innovative technique, called gendis, was proposed to extract a collection of smaller subsequences, i.e., shapelets, from a time series dataset that are very informative in classifying each of the time series into categories. gendis searches for this set of shapelets through evolutionary computation, a paradigm mostly unexplored within the domain of time series classification, which offers several benefits:evolutionary algorithms are gradient-free, allowing for an easy configuration of the optimization objective, which does not need to be differentiableonly the maximum length of all shapelets has to be tuned, as opposed to the number of shapelets and the length of each shapelet, due to the fact that gendis evaluates entire sets of shapeletseasy control over the runtime of the algorithmthe possibility of discovering shapelets that do not need to be a subsequence of the input time series

Moreover, the proposed technique has a computational complexity that is multiple orders of magnitude smaller ($O(GPKNM^{2})$ vs. $O(N^{2}M^{4})$) than the current state-of-the-art, st, while outperforming it in terms of predictive performance, with much smaller shapelet sets.

We demonstrate these benefits through intuitive experiments where it was shown that techniques that evaluate single candidates can perform subpar on imbalanced datasets and how sometimes the necessity arises to extract shapelets that are not subsequences of input time series to achieve good separation. In addition, we compare the efficiency of the different genetic operators on six different datasets and assess the algorithm’s stability by comparing the output of two different runs on the same dataset. Moreover, we conduct an extensive comparison on a large amount of datasets to show that gendis is competitive to the current state-of-the-art while having a much lower computational complexity.

Several interesting directions can be identified for further research. First, optimizations in order to decrease the required runtime for gendis can be implemented, which would allow gendis to evolve larger populations in a similar amount of time. One significant optimization is to express the calculation of distances, one of the bottlenecks of gendis, algebraically in order to leverage gpu technologies. Second, further research on each of the operators within gendis can be performed. While we clearly demonstrated the feasibility of a genetic algorithm to achieve state-of-the-art performances with the operators discussed in this work, the amount of available research within the domain of time series analysis is growing rapidly. New insights from this domain can continuously be integrated within gendis. As an example, new time series clustering algorithms could be integrated as initialization operators of the genetic algorithm. Finally, it could be very valuable to extend gendis to work with multivariate data and to discover multivariate shapelets. This would require a different representation of the individuals in the population and custom genetic operators.

## Figures and Tables

**Figure 1 sensors-21-01059-f001:**
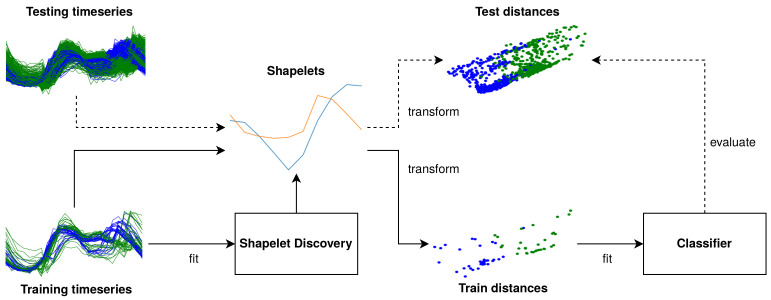
A schematic overview of shapelet discovery.

**Figure 2 sensors-21-01059-f002:**
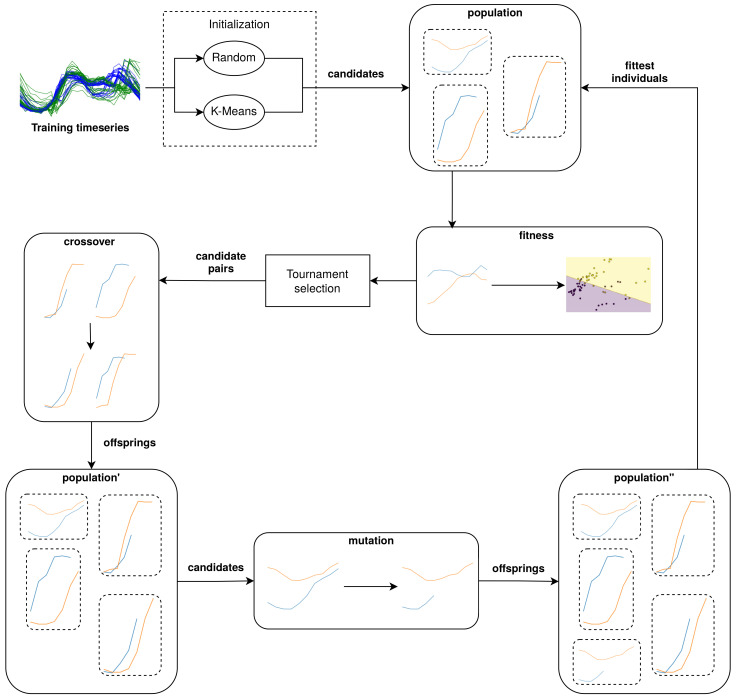
A conceptual overview of GENetic DIscovery of Shapelets (gendis).

**Figure 3 sensors-21-01059-f003:**
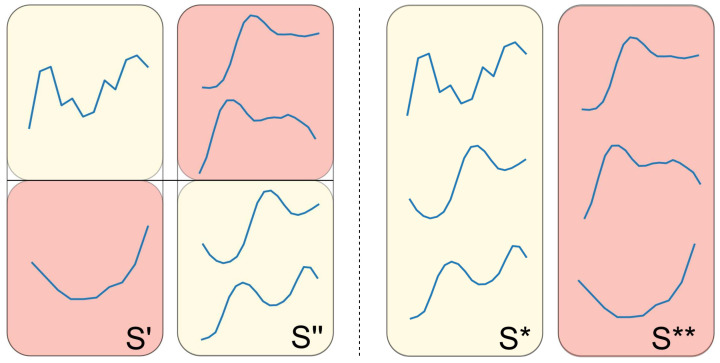
An example of a one-point crossover operation on two shapelet sets.

**Figure 4 sensors-21-01059-f004:**
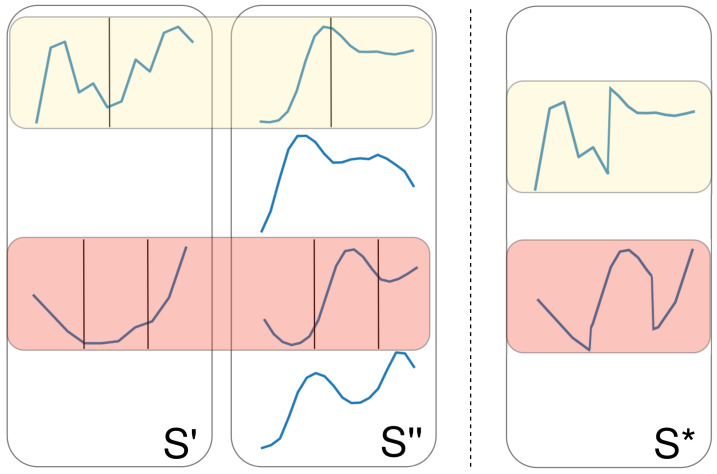
An example of one- and two-point crossover applied on individual shapelets.

**Figure 5 sensors-21-01059-f005:**
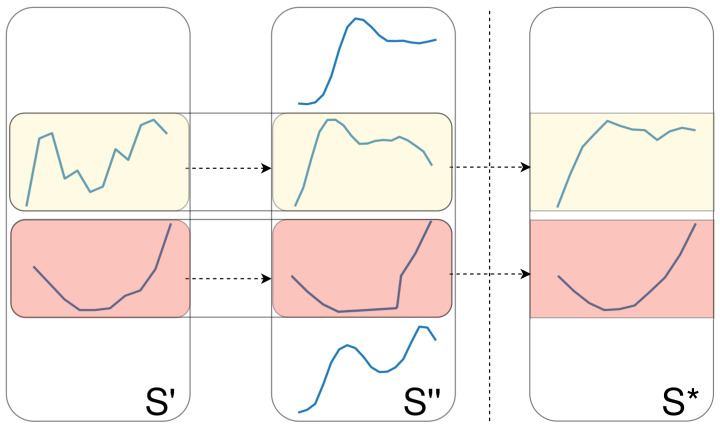
An example of the shapelet merging crossover operation.

**Figure 6 sensors-21-01059-f006:**
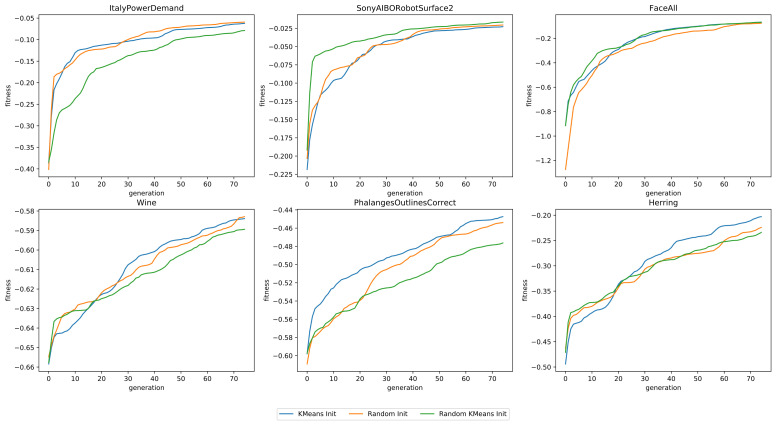
The fitness as a function of the number of generations, for six datasets, using three different configurations of initialization operations.

**Figure 7 sensors-21-01059-f007:**
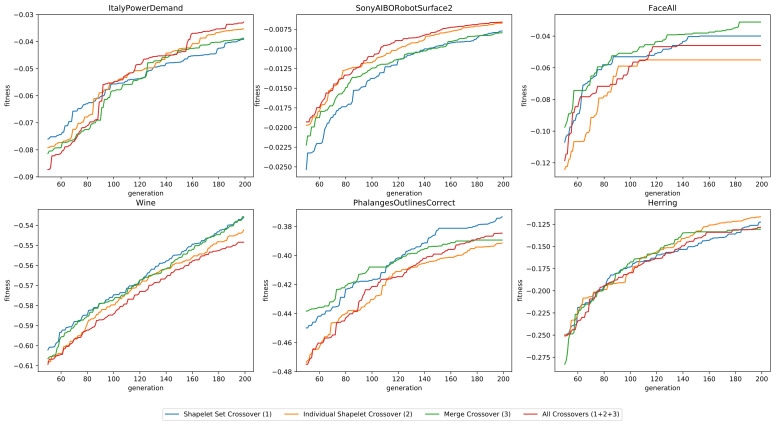
The fitness as a function of the number of generations, for six datasets, using four different configurations of crossover operations.

**Figure 8 sensors-21-01059-f008:**
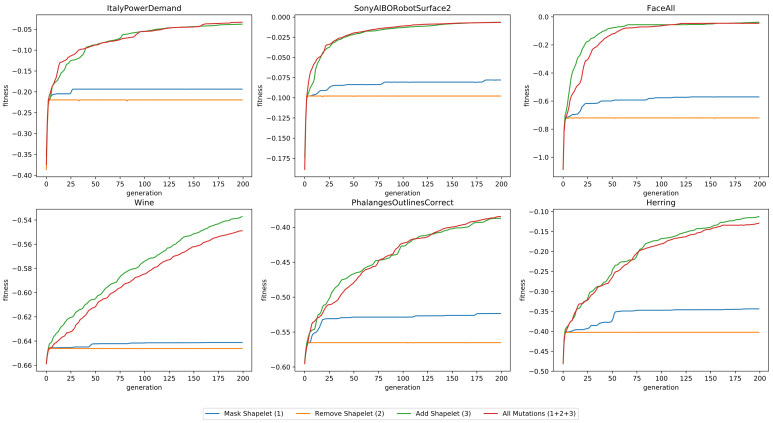
The fitness as a function of the number of generations, for six datasets, using four different configurations of mutation operations.

**Figure 9 sensors-21-01059-f009:**
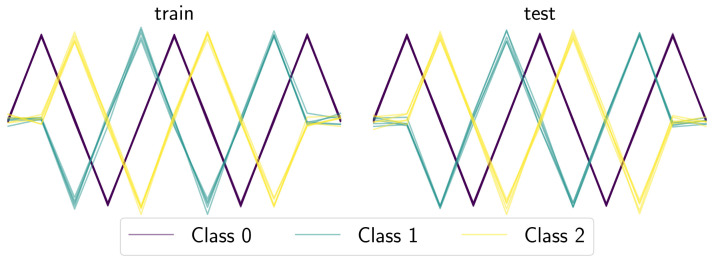
The generated training and test set for the artificial classification problem.

**Figure 10 sensors-21-01059-f010:**
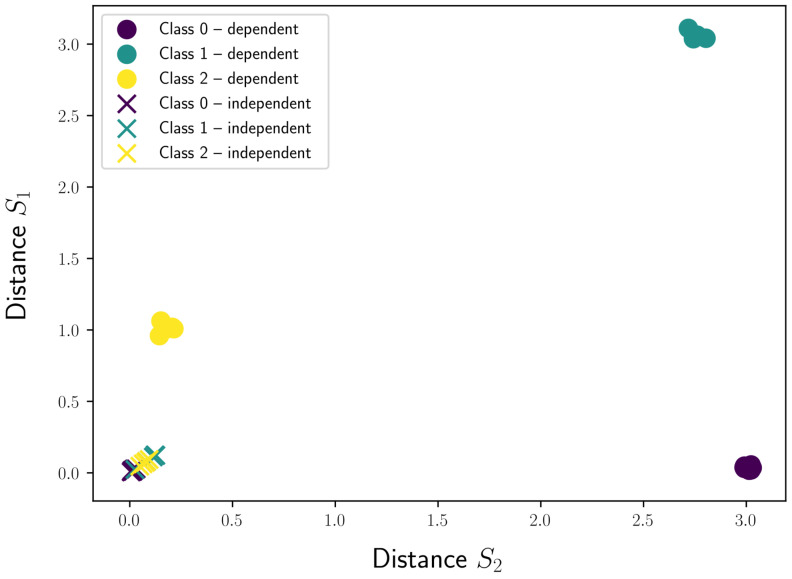
The obtained feature representations using a dependent and independent approach.

**Figure 11 sensors-21-01059-f011:**
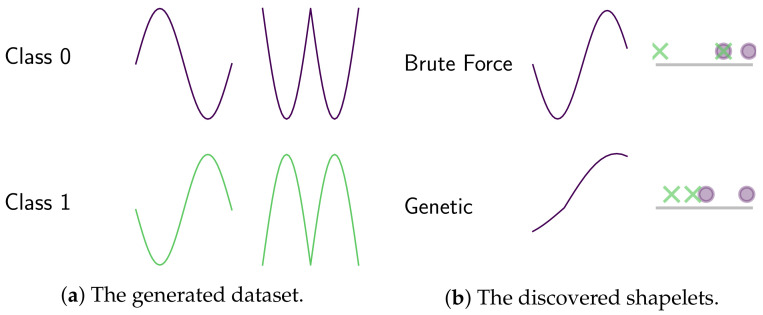
A two-class problem with two time series per class and the extracted shapelets with corresponding distances on an ordered line by a brute-force approach versus gendis.

**Figure 12 sensors-21-01059-f012:**
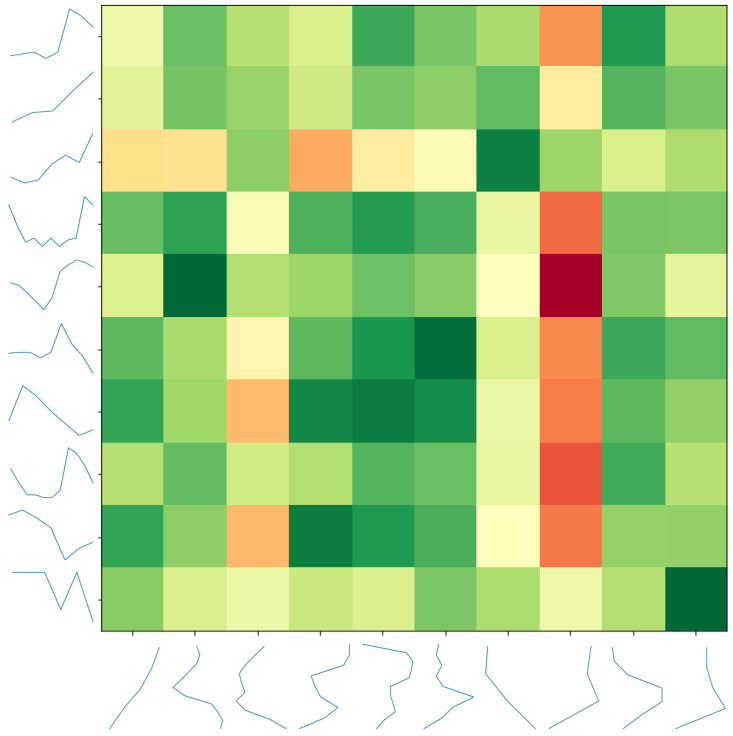
A pairwise distance matrix between discovered shapelet sets of two different runs.

**Figure 13 sensors-21-01059-f013:**
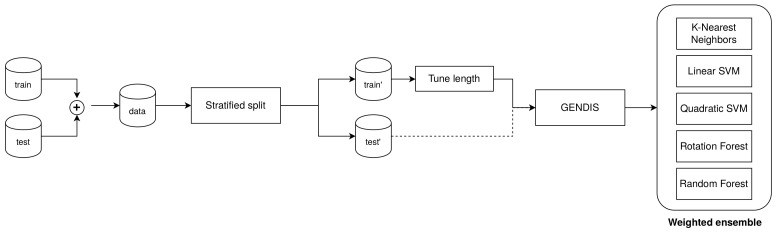
The evaluation setup used to compare gendis to other shapelet techniques.

**Figure 14 sensors-21-01059-f014:**
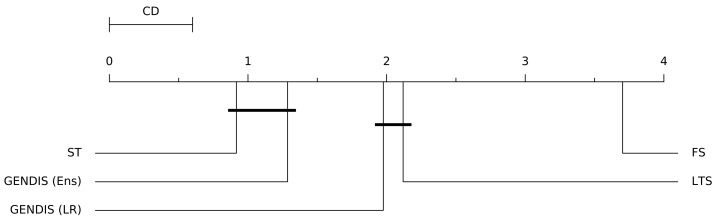
A critical difference diagram of the four evaluated shapelet discovery algorithms with a significance level of 0.1.

**Table 1 sensors-21-01059-t001:** The chosen datasets, having varying characteristics, for the evaluation of the genetic operators’ efficiency. #Cls = number of Classes, TS_len = length of Time Series, #Train = number of Training time series, #Test = number of Testing time series.

Dataset	#Cls	TS_len	#Train	#Test
ItalyPowerDemand	2	24	67	1029
SonyAIBORobotSurface2	2	65	27	953
FaceAll	14	131	560	1690
Wine	2	234	57	54
PhalangesOutlinesCorrect	2	80	1800	858
Herring	2	512	64	64

**Table 2 sensors-21-01059-t002:** A comparison between gendis and three other shapelet techniques on 85 datasets. ST, Shapelet Transform; LTS, Learning Time Series Shapelets; FS, Fast Shapelets; #Cls = number of Classes; TS_len = length of Time Series; #Train = number of Training time series; #Test = number of Testing time series; #Shapes = number of extracted shapelets by GENDIS.

Dataset	#Cls	TS_len	#Train	#Test	GENDIS		ST	LTS	FS	#Shapes
					Ens	LR				
Adiac	37	176	390	391	66.2	69.8	**76.8**	42.9	55.5	39
ArrowHead	3	251	36	175	79.4	82.0	85.1	84.1	67.5	39
Beef	5	470	30	30	51.5	58.8	**73.6**	69.8	50.2	41
BeetleFly	2	512	20	20	90.6	87.5	87.5	86.2	79.6	42
BirdChicken	2	512	20	20	90.0	90.5	92.7	86.4	86.2	45
CBF	3	128	30	900	99.1	97.6	98.6	97.7	92.4	43
Car	4	577	60	60	82.5	83.0	**90.2**	85.6	73.6	48
ChlorineConcentration	3	166	467	3840	60.9	57.5	**68.2**	58.6	56.6	30
CinCECGTorso	4	1639	40	1380	92.1	91.5	91.8	85.5	74.1	57
Coffee	2	286	28	28	98.9	98.6	99.5	99.5	91.7	44
Computers	2	720	250	250	75.4	72.7	**78.5**	65.4	50.0	38
CricketX	12	300	390	390	73.1	66.6	**77.7**	74.4	47.9	41
CricketY	12	300	390	390	69.8	64.3	**76.2**	72.6	50.9	41
CricketZ	12	300	390	390	72.6	65.9	**79.8**	75.4	46.6	40
DiatomSizeReduction	4	345	16	306	**97.4**	96.5	91.1	92.7	87.3	45
DistalPhalanxOutlineAgeGroup	3	80	400	139	**84.4**	83.2	81.9	81.0	74.5	32
DistalPhalanxOutlineCorrect	2	80	600	276	82.4	81.5	82.9	82.2	78.0	32
DistalPhalanxTW	6	80	400	139	**76.7**	76.0	69.0	65.9	62.3	33
ECG200	2	96	100	100	86.3	86.5	84.0	87.1	80.6	30
ECG5000	5	140	500	4500	94.0	93.8	**94.3**	94.0	92.2	34
ECGFiveDays	2	136	23	861	99.9	**100.0**	95.5	98.5	98.6	33
Earthquakes	2	512	322	139	**78.4**	73.7	73.7	74.2	74.7	44
ElectricDevices	7	96	8926	7711	83.7	77.6	**89.5**	70.9	26.2	31
FaceAll	14	131	560	1690	94.5	92.6	**96.8**	92.6	77.2	38
FaceFour	4	350	24	88	93.2	94.1	79.4	**95.7**	86.9	41
FacesUCR	14	131	200	2050	90.1	89.0	90.9	**93.9**	70.1	39
FiftyWords	50	270	450	455	**72.9**	71.8	71.3	69.4	51.2	39
Fish	7	463	175	175	87.0	90.5	**97.4**	94.0	74.2	50
FordA	2	500	3601	1320	90.8	90.7	**96.5**	89.5	78.5	37
FordB	2	500	3636	810	89.5	89.8	**91.5**	89.0	78.3	38
GunPoint	2	150	50	150	96.9	95.7	**99.9**	98.3	93.0	39
Ham	2	431	109	105	72.9	77.2	80.8	**83.2**	67.7	37
HandOutlines	2	2709	1000	370	89.7	91.0	**92.4**	83.7	84.1	41
Haptics	5	1092	155	308	45.2	43.9	**51.2**	47.8	35.6	55
Herring	2	512	64	64	59.6	61.8	**65.3**	62.8	55.8	42
InlineSkate	7	1882	100	550	**43.8**	39.3	39.3	29.9	25.7	58
InsectWingbeatSound	11	256	220	1980	57.3	57.5	**61.7**	55.0	48.8	36
ItalyPowerDemand	2	24	67	1029	95.6	**96.0**	95.3	95.2	90.9	31
LargeKitchenAppliances	3	720	375	375	91.0	90.4	**93.3**	76.5	41.9	33
Lightning2	2	637	60	61	**80.9**	79.1	65.9	75.9	48.0	39
Lightning7	7	319	70	73	**78.2**	76.3	72.4	76.5	10.1	39
Mallat	8	1024	55	2345	**98.2**	97.3	97.2	95.1	89.3	58

**Table 3 sensors-21-01059-t003:** A comparison between gendis and three other shapelet techniques on 85 datasets. ST, Shapelet Transform; LTS,Learning Time Series Shapelets; FS, Fast Shapelets; #Cls = number of Classes; TS_len = length of Time Series; #Train = number of Training time series; #Test = number of Testing time series; #Shapes = number of extracted shapelets by GENDIS.

Dataset	#Cls	TS_len	#Train	#Test	GENDIS		ST	LTS	FS	#Shapes
					Ens	LR				
Meat	3	448	60	60	98.7	**98.8**	96.6	81.4	92.4	48
MedicalImages	10	99	381	760	**72.4**	68.6	69.1	70.4	60.9	37
MiddlePhalanxOutlineAgeGroup	3	80	400	154	**74.4**	73.2	69.4	67.9	61.3	30
MiddlePhalanxOutlineCorrect	2	80	600	291	80.7	79.6	81.5	**82.2**	71.6	30
MiddlePhalanxTW	6	80	399	154	62.2	**63.1**	57.9	54.0	51.9	37
MoteStrain	2	84	20	1252	86.3	86.6	88.2	87.6	79.3	36
NonInvasiveFatalECGThorax1	42	750	1800	1965	84.7	89.4	**94.7**	60.0	71.0	41
NonInvasiveFatalECGThorax2	42	750	1800	1965	87.1	92.3	**95.4**	73.9	75.8	37
OSULeaf	6	427	200	242	76.2	75.8	**93.4**	77.1	67.9	45
OliveOil	4	570	30	30	86.1	88.8	88.1	17.2	76.5	53
PhalangesOutlinesCorrect	2	80	1800	858	**80.8**	78.7	79.4	78.3	73.0	30
Plane	7	144	105	105	99.2	99.3	**100.0**	99.5	97.0	34
ProximalPhalanxOutlineAgeGroup	3	80	400	205	84.1	83.9	84.1	83.2	79.7	32
ProximalPhalanxOutlineCorrect	2	80	600	291	86.2	85.7	**88.1**	79.3	79.7	30
ProximalPhalanxTW	6	80	400	205	**81.7**	80.2	80.3	79.4	71.6	31
RefrigerationDevices	3	720	375	375	68.6	62.4	**76.1**	64.2	57.4	34
ScreenType	3	720	375	375	52.8	52.8	**67.6**	44.5	36.5	37
ShapeletSim	2	500	20	180	**100.0**	100.0	93.4	93.3	100.0	34
ShapesAll	60	512	600	600	79.6	79.3	**85.4**	76.0	59.8	44
SmallKitchenAppliances	3	720	375	375	74.3	74.2	**80.2**	66.3	33.3	37
SonyAIBORobotSurface1	2	70	20	601	**96.0**	95.6	88.8	90.6	91.8	34
SonyAIBORobotSurface2	2	65	27	953	88.6	87.0	**92.4**	90.0	84.9	34
StarLightCurves	3	1024	1000	8236	95.9	95.3	**97.7**	88.8	90.8	30
Strawberry	2	235	613	370	95.2	95.1	**96.8**	92.5	91.7	34
SwedishLeaf	15	128	500	625	88.7	87.7	**93.9**	89.9	75.8	37
Symbols	6	398	25	995	**93.4**	92.5	86.2	91.9	90.8	43
SyntheticControl	6	60	300	300	98.8	98.7	98.7	**99.5**	92.0	39
ToeSegmentation1	2	277	40	228	92.0	90.7	**95.4**	93.4	90.4	40
ToeSegmentation2	2	343	36	130	93.1	90.6	94.7	94.3	87.3	36
Trace	4	275	100	100	100.0	99.9	100.0	99.6	99.8	25
TwoLeadECG	2	82	23	1139	95.8	96.6	98.4	**99.4**	92.0	36
TwoPatterns	4	128	1000	4000	95.8	93.1	95.2	**99.4**	69.6	33
UWaveGestureLibraryAll	8	945	896	3582	**94.9**	94.8	94.2	68.0	76.6	44
UWaveGestureLibraryX	8	315	896	3582	80.2	77.9	80.6	80.4	69.4	42
UWaveGestureLibraryY	8	315	896	3582	71.5	69.5	**73.7**	71.8	59.1	40
UWaveGestureLibraryZ	8	315	896	3582	74.4	72.2	74.7	73.7	63.8	42
Wafer	2	152	1000	6164	99.4	99.3	**100.0**	99.6	98.1	24
Wine	2	234	57	54	86.5	86.4	**92.6**	52.4	79.4	39
WordSynonyms	25	270	267	638	**68.4**	62.8	58.2	58.1	46.1	42
Worms	5	900	181	77	65.6	59.9	**71.9**	64.2	62.2	47
WormsTwoClass	2	900	181	77	74.1	69.1	**77.9**	73.6	70.6	50
Yoga	2	426	300	3000	83.3	80.2	82.3	83.3	72.1	40

## Data Availability

An implementation of gendis in Python 3 is available on GitHub (https://github.com/IBCNServices/GENDIS). Moreover, the code in order to perform the experiments to reproduce the results is included.

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
