# Peer review of "GENDIS: Genetic Discovery of Shapelets"

_sensors, 2021, doi:10.3390/s21041059_

Round 1
Reviewer 1 Report
This manuscript proposed an innovative approach to extract shapelets from time-series data. The proposed method has smaller computational complexity than other state-of-the-art methods. Finally, the experimental investigation was conducted to validate the effectiveness of the proposed technique. Overall, the topic of this research is interesting, and the manuscript was well organized and written. I suggest that it can be considered to be published in Sensors if the authors can well address the following comments.
- Please illustrate the main innovation of this study. There are a lot of similar techniques. Why was this method proposed for the task of interest?
- A flowchart is necessary for better demonstrating the proposed method.
- In addition to the calculation complexity, are there other indicators for the method comparison?
- Future research should be included in conclusion part.
Author Response
This manuscript proposed an innovative approach to extract shapelets from time-series data. The proposed method has smaller computational complexity than other state-of-the-art methods. Finally, the experimental investigation was conducted to validate the effectiveness of the proposed technique. Overall, the topic of this research is interesting, and the manuscript was well organized and written. I suggest that it can be considered to be published in Sensors if the authors can well address the following comments.
- Please illustrate the main innovation of this study. There are a lot of similar techniques. Why was this method proposed for the task of interest?
We added the main innovation of this study as the first line of the section “Our Contribution” such that it is well highlighted. The sentence reads: “This paper is the first to investigate the feasibility of an evolutionary algorithm in order to discover a set of shapelets from a collection of labeled time series.” - A flowchart is necessary for better demonstrating the proposed method.
We have added a new subsection with a conceptual overview (flowchart) of GENDIS (subsection 2.5.1). - In addition to the calculation complexity, are there other indicators for the method comparison?
We believe predictive performance to be the most important dimension to compare techniques along, next to scalability. We therefore spent most of our effort on performing a thorough comparison on a large number of dataset. - Future research should be included in conclusion part.
We have renamed the section “Conclusions” to “Conclusion and future work” and extended it with several possible future research directions.
Reviewer 2 Report
MDPI Sensors: Reviewer’s comments to the paper:
GENDIS: GENetic DIscovery of Shapelets
The study seems interesting and useful in the corresponding research field.
The paper, in general, satisfies major rigor requirements that are demanded from MDPI Sensors. The enormous research efforts seem to be evident. The red clue remains consistent throughout the paper, while the main contributions and findings are clearly emphasized. Moreover, in general, the paper is well organized and written. So, to summarize, the submitted paper's contribution seems relevant and useful for the researchers from the field and should likely have earned an opportunity to be accepted and published in the journal.
The reviewer thinks that only one minor issue should be fixed:
- Please, add a new section titled “The conceptual framework” that should be included somewhere between the introduction section and methods section. In this section, please give a certain block diagram or flow chart (or pseudo-code) of the consecutive steps that have been carried out in this research. In this block diagram, it must be clearly seen what the main inputs and outputs between the consecutive steps of the entire new mechanism are. This way, the reader will get a chance to easier follow a further text, which is then explained step by step.
Author Response
The study seems interesting and useful in the corresponding research field.
The paper, in general, satisfies major rigor requirements that are demanded from MDPI Sensors. The enormous research efforts seem to be evident. The red clue remains consistent throughout the paper, while the main contributions and findings are clearly emphasized. Moreover, in general, the paper is well organized and written. So, to summarize, the submitted paper's contribution seems relevant and useful for the researchers from the field and should likely have earned an opportunity to be accepted and published in the journal.
The reviewer thinks that only one minor issue should be fixed:
- Please, add a new section titled “The conceptual framework” that should be included somewhere between the introduction section and methods section. In this section, please give a certain block diagram or flow chart (or pseudo-code) of the consecutive steps that have been carried out in this research. In this block diagram, it must be clearly seen what the main inputs and outputs between the consecutive steps of the entire new mechanism are. This way, the reader will get a chance to easier follow a further text, which is then explained step by step.
We have added a new subsection with a conceptual overview (flowchart) of GENDIS (subsection 2.5.1) right before the sections that provide more details on each of the operators.
Reviewer 3 Report
Thank you very much for the opportunity to review this very interesting article, which certainly has great potential. I did not see any major shortcomings. The shortcomings are more of a formal nature (see below). I appreciate an interesting Introduction. More literature sources could be cited in the "Related Work" section. Literature review is not enough, the latest quality sources are missing. The part devoted to the "Materials and Methods" used is sufficiently specific, for completeness the formulas should be numbered. In the title of Figure 1 you have a description, which should be rather a separate text or a Note. It should not be part of the figure title. The same for Figure 2, 10, 12, 13 or for Tables 1, 2, 3.
In the "Results" section, you describe the datasets at the beginning. Rather, this section should be part of section "2. Materials and Methods". The "Discussion" section is missing. Some links and comparisons with other authors are the content of the results section. If you leave it like this, I recommend calling the chapter "Results and Discussion". "Conclusions" is perhaps the weakest part of your article. There are no research limits and no proposal for future research. It would also be good to add information on where specifically in practice the proposed innovative gendis technique can be used. There are minor formal shortcomings in the "References", I recommend checking.
Author Response
Thank you very much for the opportunity to review this very interesting article, which certainly has great potential. I did not see any major shortcomings. The shortcomings are more of a formal nature (see below). I appreciate an interesting Introduction.
- More literature sources could be cited in the "Related Work" section. Literature review is not enough, the latest quality sources are missing.
We have added two new paragraphs with related work: one that discusses fundamental extensions to shapelet discovery and one paragraph that discusses extensions of existing algorithms for multi-variate time series. - The part devoted to the "Materials and Methods" used is sufficiently specific, for completeness the formulas should be numbered.
We have numbered all equations. - In the title of Figure 1 you have a description, which should be rather a separate text or a Note. It should not be part of the figure title. The same for Figure 2, 10, 12, 13 or for Tables 1, 2, 3.
We have shortened all the captions of the figures & tables. - In the "Results" section, you describe the datasets at the beginning. Rather, this section should be part of section "2. Materials and Methods".
The datasets discussed in Section 3.1 are specific to the results presented in that Section and are a subset of the 85 datasets used in a further section for performance comparison to other algorithms. We therefore believe that the discussion of these datasets is better suited in the “Results” section. - The "Discussion" section is missing. Some links and comparisons with other authors are the content of the results section. If you leave it like this, I recommend calling the chapter "Results and Discussion".
We have followed your suggestion and renamed the section to "Results and Discussion", we believe the interweaving of results and discussion is easier to follow for the reader than two separate sections. - "Conclusions" is perhaps the weakest part of your article. There are no research limits and no proposal for future research. It would also be good to add information on where specifically in practice the proposed innovative gendis technique can be used.
We have renamed the section “Conclusions” to “Conclusion and future work” and extended it with several possible future research directions which also highlight some of limitations of the work presented in this study. - There are minor formal shortcomings in the "References", I recommend checking.
We have double-checked all our references to solve all formatting issues and to complete missing information.